# A Bioinformatics View of Glycan–Virus Interactions

**DOI:** 10.3390/v11040374

**Published:** 2019-04-23

**Authors:** Philippe Le Mercier, Julien Mariethoz, Josefina Lascano-Maillard, François Bonnardel, Anne Imberty, Sylvie Ricard-Blum, Frédérique Lisacek

**Affiliations:** 1Swiss-Prot Group, SIB Swiss Institute of Bioinformatics, 1211 Geneva, Switzerland; 2Proteome Informatics Group, SIB Swiss Institute of Bioinformatics, 1211 Geneva, Switzerland; julien.mariethoz@sib.swiss (J.M.); Josefina.LascanoMaillard@unige.ch (J.L.-M.); francois.bonnardel@cermav.cnrs.fr (F.B.); 3Computer Science Department, University of Geneva, 1227 Geneva, Switzerland; 4Université Grenoble Alpes, CNRS, CERMAV, 38041 Grenoble, France; anne.imberty@cermav.cnrs.fr; 5Institut de Chimie et Biochimie Moléculaires et Supramoléculaires, UMR 5246 CNRS—Université Lyon 1, 69622 Villeurbanne Cedex, France; sylvie.ricard-blum@univ-lyon1.fr; 6Section of Biology, University of Geneva, 1211 Geneva, Switzerland

**Keywords:** glycan, carbohydrate, bioinformatics, host–virus interactions, immunology, database, data integration, interoperability

## Abstract

Evidence of the mediation of glycan molecules in the interaction between viruses and their hosts is accumulating and is now partially reflected in several online databases. Bioinformatics provides convenient and efficient means of searching, visualizing, comparing, and sometimes predicting, interactions in numerous and diverse molecular biology applications related to the -omics fields. As viromics is gaining momentum, bioinformatics support is increasingly needed. We propose a survey of the current resources for searching, visualizing, comparing, and possibly predicting host–virus interactions that integrate the presence and role of glycans. To the best of our knowledge, we have mapped the specialized and general-purpose databases with the appropriate focus. With an illustration of their potential usage, we also discuss the strong and weak points of the current bioinformatics landscape in the context of understanding viral infection and the immune response to it.

## 1. Introduction

Viruses are genetic mobile elements that replicate into a host cell and produce viral particles to infect new cells or organisms. The virion journey in the extracellular space is an opportunity to interact in many ways with one of the most abundant molecules in that area: glycans. These multifunctional molecules cover most of epithelial or bacterial cells in a thick layer, called glycocalyx [1], or as soluble macromolecules called mucins [2]. In turn, numerous proteins are expressed at the surface of eukaryotic cells or viruses that bind those glycans. The glycocalyx can be a barrier for viruses that target cellular membranes [3,4], but also an opportunity to bind specific targets for virus replication [5].

These host–virus interactions are dynamic and multifaceted. Some viruses have evolved to bind specific glycans in order to infect cells, like human rotaviruses binding to blood group A antigens [6]. Infecting cells has consequences, and the immune system is able to recognize some virions through their glycans. This is the case of macrophages or dendritic cells that phagocyte virions through binding glycans at their surface as observed in HIV (Human Immunodeficiency Virus) [7], Ebola virus [8], HCV (Hepatitis C virus) [9], as well as the influenza [10] or SARS (Severe Acute Respiratory Syndrome) viruses [11]. In turn, this anti-viral system is exploited by several viruses that target macrophages or dendritic cells for their replication, like the Ebola virus [8] and SARS virus [11]. Another unexpected role of host glycans is to evade immune system recognition of viral molecules. By having their virion surface proteins covered with host glycans, specific viral epitopes are physically hidden under neutral host molecules and cannot be recognized by immune cells. This effect is referred as to the glycan shield and the best example of this feature are the HIV virus, known to have the highest density of glycans attached to its surface proteins [12], and the Lassa virus [13]. 

All of these functions are critical for viral replication of most pathogenic viruses and could be exploited to create antiviral compounds [14]. Nonetheless this remains a challenge because of the complexity of these interactions involving diverse host–virus hybrid molecules. The cellular machinery produces numerous sugars that can be combined into a broad variety of glycosylation types, allowing an extended heterogeneity of virion surface molecules. Studying such multifaceted interplay between glycans and viruses results in complex data that cannot be thoroughly understood without databases and bioinformatics [15]. The interpretation of large-scale analyses depends on comparative and predictive approaches in which biological databases play a crucial role. This emphasizes the importance of collecting, storing, curating, and making these data searchable online. Next, crosslinking the various specialized databases brings the user a broader view to grasp the complexity of multi-organism interactions. In the present review, we suggest that bioinformatics and databases are key to harnessing the potential for new knowledge discovery in virus and vaccine research. We also show that integrating complementary information from multiple sources is not straightforward.

## 2. The Current Landscape of Bioinformatics Resources for Molecular Glycovirology

### 2.1. Overview

A handful of bioinformatics resources include the relevant pieces of information that can be put together to study glycan–virus interactions at different stages of infection. To follow the outline of a recent review highlighting the roles of glycosylation in virus biology [16], we have mapped the corresponding databases as shown in Figure 1. Overall, these roles pertain to immunity, attachment, entry, and exit, as well as glycoprotein synthesis. 

ViralZone [17] is a web resource that bridges sequence data to virus knowledge thereby providing the means of exploring the diversity of viruses. It links to viral proteome sequences contained in UniProtKB [18], the Universal Protein Knowledgebase. UniProtKB is also a reliable source for describing host glycoproteins that are further characterized in neXtProt [19] when the host is human. Irrespective of their origin, knowledge of glycoproteins and the glycans they carry is stored in GlyConnect [20].

In fact, most viruses use the host glycosylation machinery composed of enzymes that are categorized in the CAZy database of Carbohydrate-Active enZymes [21] upon sequence similarity and biosynthetic or degrading activity. Glycans are the substrates and products of these reactions. GlyTouCan [22] is the most comprehensive glycan structure repository.

ViralZone, includes detailed information on virus taxonomy, virion structures, as well as their interactions with other molecules. This latter aspect highlights the frequent occurrence of glycan mediation in receptor binding. This is shown, for example, with cross-references to SugarBindDB [23] that collects information regarding host surface glycan recognition by pathogenic agents such as viruses. Other instances of viral protein recognition via heparan sulfate proteoglycan (co)-receptors are cited in ViralZone though do not yet referr to MatrixDB [24], a database that maps experimentally proven interactions between ECM (extracellular matrix) proteins, proteoglycans, and glycosaminoglycans (GAGs). More generally, carbohydrate-binding proteins of all origins are compiled and classified in UniLectin [25], a recently released platform for the study of lectins.

Major molecules involved in immunity (antibodies and antigens) are extensively covered in two very popular and long-standing databases, namely: the International Immuno-Genetics information system (IMGT) [26] and the Immune Epitope Database (IEDB) [27]. More recently, a Database of Anti-Glycan Reagents (DAGR) [28] focused on anti-glycan antibodies was launched. Unfortunately, DAGR is currently devoid of any cross-reference to existing databases. In fact, the obvious advantage of mapping the resources in relation to virus cycle as shown in Figure 1, is to emphasize the potential for exploring knowledge via cross-references. The connectivity of databases is shown in Figure 2 where the renowned UniProtKB and Protein Data Bank (PDB) [29] obviously play a central role. As such, they provide general knowledge that can then be refined through consulting the more specialized resources featured in the figure. The content description and relevance to virology of each resource featuring in Figure 2 are provided in Table 1. The creation date was added as an indication of not only the reliability of sources, such as UniProtKB, PDB and IMGT, but also the profusion of recent effort in glycoinformatics.

Note that each and every of these resources is pointing to PubMed [30], the biomedical literature search engine that provides access to bibliographic information. This systematic link is not pictured to avoid crowding the figure with arrows, nonetheless the existence of shared PubMed IDs between databases is an asset for knowledge mining. We illustrate this point with PubMed ID 12825167 [31] that is common between UniProtKB and SugarBindDB. A scenario of navigation across databases is shown in Figure 3.

**Table 1 viruses-11-00374-t001:** The databases pictured in Figure 2 are briefly described and qualified for their relevance to virology.

Name	URL	Main Content	Relevance to Virology	Example of Use	Data Status	Created in	Reference
**ViralZone**	viralzone.expasy.org	Illustrated encyclopedia of viruses	Links textbook knowledge to sequence data	Explore the specific molecular biology of a virus	Curated	2008	[17]
**UniProt**	www.uniprot.org	Integrated knowledge of proteins	Provides protein sequence and functional information.	Find glycosylation sites in protein sequence	Curated (Swiss-Prot section)	1986	[18]
**neXtprot**	www.nextprot.org	Integrated knowledge of human proteins	Provides detailed functional information on human proteins	Characterize human host surface receptors	Partially curated	2009	[19]
**PDBe**	www.ebi.ac.uk/pdbe	3D structures of proteins	Structural virology: capsid structures, host–virus interactions.	Examine the 3D structure of a capsid protein	Partially curated	1980	[29]
**ChEBI**	www.ebi.ac.uk/chebi	Information on chemical compounds	Describes chemical compounds interacting with viruses	Explore the molecule formula of a given compound	Curated	2004	[32]
**PubChem**	pubchem.ncbi.nlm.nih.gov	Information on chemical compounds and corresponding assays	Collects assays involving chemical compounds interacting with viruses	Find binding assays associated with a virus	Minimally curated	2004	[33]
**CAZy**	www.cazy.org	Carbohydrate Active Enzymes (CAZymes)	Families of viral CAZymes	Find viral CAZymes and associated Enzyme Nomenclature numbers	Curated	1998	[21]
**GlyTouCan**	www.glytoucan.org	Glycan 2D structures	Provides information on glycan structure.	Check existence of glycan bound by virus	Not curated	2016	[22]
**Rhea**	www.rhea.org	Enzymatic reactions	Describes virus specific enzyme reactions	Find substrate and product of CAZyme reaction	Curated	2011	[34]
**MatrixDB**	matrixdb.univ-lyon1.fr	Extracellular matrix components and interactions	Describes glycosaminoglycans and their interactions	Find interactions involving heparan sulfate	Curated	2008	[24]
**SugarBind**	sugarbind.expasy.org	Host glycans and pathogen lectins	Describes glycan–virus interactions	Explore pathogen lectins/adhesins binding a glycan motif	Curated	2005	[23]
**UniLectin3D**	www.unilectin.eu	Carbohydrate-binding proteins (not Ab)	Collects glycan–virus interactions	Inspect atomic details of glycan-haemagglutinin interactions	Curated	2018	[25]
**GlyConnect**	glyconnect.expasy.org	Integrated knowledge of glycoproteins	Describes glycoproteins	Find sialylated glycans on receptor proteins	Curated	2017	[20]
**IMGT**	www.imgt.org	Integrated knowledge of immunoglobulins	Collects antibody sequences and structures	Find antiviral antibody sequences	Curated	1989	[26]
**IEDB**	www.iedb.org	Collected knowledge of antigenic ligands (epitopes)	Collect experimental data characterizing the antigenicity of a virus.	Find viral antigenic peptides	Curated	2003	[27]
**DAGR**	ccr2.cancer.gov/resources/Cbl/Tools/Antibody	Carbohydrate-binding antibodies	Collects anti-glycan reagents	Find glycan binding motifs of antiviral antibodies	Curated	2016	[28]

**Table 2 viruses-11-00374-t002:** The tools either used in the example of Figure 3 or mentioned in the text as known to visualize or analyze glycan-binding data are briefly described and qualified for their relevance to virology.

Name	URL	Purpose	Applicable to Virology	Created in	Reference
**LiteMol**	www.litemol.org	Visualization of protein 3D structure	yes	2015	[35]
**GlyS^3^**	glycoproteome.expasy.org/substructuresearch	Glycan substructure search	yes	2015	[36]
**GlycoPattern**	glycopattern.emory.edu	Analysis of glycan array data	yes	2014	[37]
**GLAD**	glycotoolkit.com/GLAD	Visualization and analysis of glycan array data	yes	2019	[38]
**Glycome Atlas**	rings.t.soka.ac.jp/GlycomeAtlasV5/index.html	Visualization of glycan array data	no	2012	[39]

**Figure 3 viruses-11-00374-f003:**
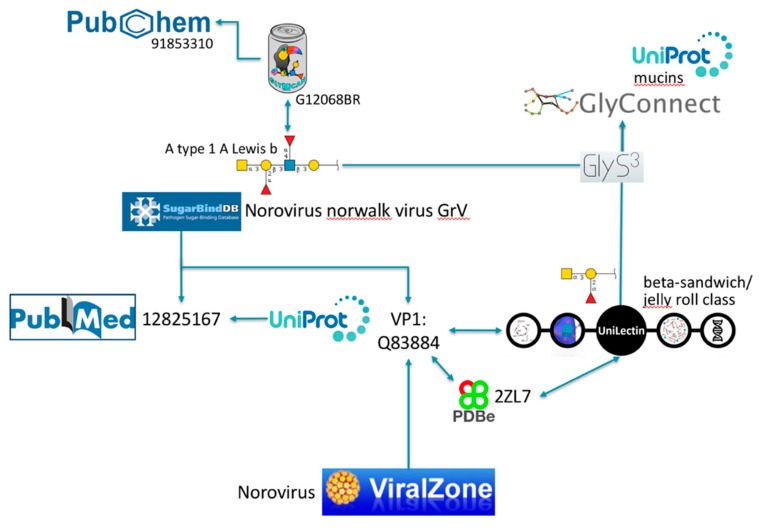
Example of exploration via cross-references.

### 2.2. Illustration of Usage

The SugarBindDB entry for Norovirus Norwalk virus GrV (the official designation should be GV according to ViralZone but SugarBindDB reports the name found in the cited source) is associated with several glycan binders one of which being A type 1 A Lewis b (GalNAc(α1–3)[Fuc(α1–2)]Gal(β1–3)[Fuc(α1–4)] GlcNAc(β1–3)Gal), characterized in PubMed ID 12825167 [31]. This article is cited as well in the cross-referenced Norwalk virus capsid protein VP1 entry of UniProtKB (Q83884). Reciprocal links between SugarBindDB and ViralZone, as well as the common UniProtKB entry, provide alternative accesses to the general knowledge on Noroviruses described in ViralZone. This very same UniProtKB protein entry is also shared by UniLectin that classifies this carbohydrate-binding protein in the beta-sandwich/jelly roll category. Other viral lectins with this fold can be explored from there. UniProtKB and UniLectin both cross-link to PDB entries associated with capsid protein VP1, but the latter database favors PDB references with bound glycan ligands. This is the case of PDB 2ZL7 that details GalNAc(α1–3)[Fuc(α1–2)]Gal as a ligand. Note that PDB is preferred as it uses the LiteMol software [35] for visualizing 3D structures. LiteMol has a convenient app for showing glycan in 3D [40], consistent with the Symbol Nomenclature For Glycans (SNFG) notation [41] that is used in Figure 3 and in all glyco-related resources cited. Both glycan ligands proposed either in SugarBindDB or in UniLectin are matched to GlyConnect using GlyS^3^, an in-house substructure search engine [36] also available as a standalone tool from the Glycomics@ExPASy collection [42]. The result is a list of full glycan structures containing the ligand substructures. Most of these full structures are attached to mucins, some of which are linked to UniProtKB.

Another course of exploration can be followed through the reciprocal link between SugarBindDB and GlyTouCan. A type 1 A Lewis b is registered in GlyTouCan and the corresponding record includes a link to molecules in PubChem [33] which, as part of the large NCBI (National Center for Biotechnology Information) collection [30], opens many further lines of investigation. 

This example illustrates the potential for identifying protein interacting partners along with details of the glycan structures and glycan ligands that mediate this interaction. If databases were to be more complete, automated procedures could rapidly extract the relevant pieces of information. At this stage, the proposed scenario can only be performed by a user clicking on available links.

## 3. Repertoire Mapping to Enrich Databases

A large part of the knowledge regarding glycan–virus interactions remains in the literature which is screened by database curators to be made available online. This is a slow process which mainly explains the current lacks in databases. Furthermore, this incomplete picture leads to focus on general-purpose resources such as UniProtKB or the PDB with PubMed in the background. Figure 2, where most arrows are concentrated on these resources, illustrates this point conclusively, as well as highlights the limited potential for glycan data integration in the immunological databases.

Substantial knowledge is also contained in the results of glycan arrays that have been used to study virus–host interactions for close to two decades. For instance, this approach revealed the subtle differences that distinguish avian from human flu based on glycan recognition by hemagglutinin [43]. There are a few online sources of data that have stored the results of glycan array screening. The one that has been considered for many years as the reference was the first online repository put together by the US-based Consortium for Functional Glycomics (CFG) [44]. However, due to the discontinuation of funding, it is no longer updated and only temporarily maintained. These data are still available for download on the CFG website (http://www.functionalglycomics.org/glycomics/publicdata/primaryscreen.jsp). Dedicated tools were also developed specifically for visualizing [39] or mining [37] the CFG array data. While the latter is also no longer maintained, the former does not include virus data. At this point in time, US-based glycan array data collecting and mining is in transition between old and new initiatives. As we write, a new platform designed to visualize, analyze and compare glycan microarray data is briefly introduced in a just released Application Note [38]. This shows that bioinformatics applied to processing and analyzing glycan array data is still very much in the making. Another reference library is hosted in the Glycosciences Laboratory of Imperial College, London. It contains many virus screens (see Category D in https://glycosciences.med.ic.ac.uk/data.html). However, each experimental result displayed from the database does not provide any cross-reference other than a link of a cited published article to PubMed.

Table 2 summarizes information on the software tools that have been cited in the text so far, mainly for visualizing structural or glycan-binding data.

### 3.1. Glycans as Virus Receptors in ViralZone

Viruses initiate infection of a host cell by attaching to specific receptors. Binding to an entry receptor triggers a chain of dynamic events that will enable cell penetration by the virus. Many virus–host receptor interactions have been described, and those are reported in the ViralZone virus receptors table (https://viralzone.expasy.org/5356) which contains 254 expert curated interactions. An excerpt of this table is shown in Figure 4. Of those, 53 are interactions with glycans (carbohydrates) from 49 different viruses. All proteinaceous molecules are linked to UniProtKB, glycans are linked to small molecule entries of ChEBI [32] through Rhea, the biochemical reaction database [34] and to SugarBindDB. Representing glycan ligands bound by the influenza virus is challenging because it binds the sialic acid cap of those, and therefore can attach to many different molecules. The link to SugarBindDB provides a clean overview with the detailed structure of 92 agents bound by the influenza virus. On the other hand, ChEBI links provide the chemical structure of involved sialic acids. Bridging these resources allows investigators to browse freely toward different aspects of biology and chemistry and constitutes a substantial added value to the data.

**Figure 4 viruses-11-00374-f004:**
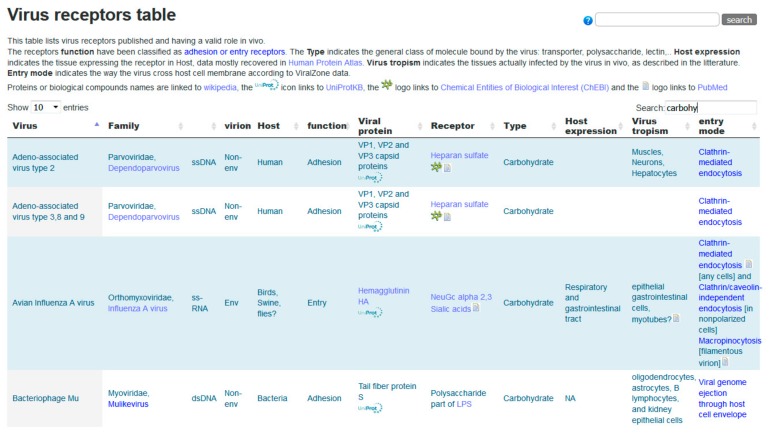
Excerpt of the ViralZone virus receptor table (https://viralzone.expasy.org/5356). The screenshot displays the first four carbohydrate receptors of viruses.

### 3.2. Emerging Knowledge in MatrixDB

The majority of interactions stored in MatrixDB involve proteins and proteoglycans and glycosaminoglycans. Numerous viruses (e.g., coronavirus, human papillomavirus virus, hepatitis B and C viruses, HIV, and Zika virus) use GAGs displayed at the surface of host cells, mostly heparan sulfate, for host cell attachment and invasion [45,46,47,48]. However, only one curated paper describes a GAG–virus receptor interaction between heparin and the Gibbon ape leukemia virus receptor 2 or sodium-dependent phosphate transporter 2 (Q08357) in MatrixDB, emphasizing the urgent need of curation. The corresponding UniProt entry states that “it functions as a retroviral receptor and confers human cells susceptibility to infection to amphotropic murine leukemia virus (A-MuLV), 10A1 murine leukemia virus (10A1 MLV) and some feline leukemia virus subgroup B (FeLV-B) variants” as supported by [49].

The field of glycosaminoglycanomics is budding, but data collection is still limited by the lack of standards to encode structural information [50]. Progress in complying with standards is the first step to increasing connectivity [51]. Furthermore, links between ChEBI and MatrixDB help bridging with chemical biology databases. GAG entries of MatrixDB are also cross-referenced with GlyTouCan.

MatrixDB is designed and built to display interaction networks that provide a snapshot of direct and indirect physical interactions between proteins, GAGs, and small molecules with a particular focus on the precise definition of extracellular matrix multimers. Ultimately, the table shown in Figure 4 will be refined to include links to MatrixDB, which in turn will share data with SugarBindDB.

### 3.3. SugarBindDB and UniLectin as Key Connectors

As visible in Figure 3, SugarBindDB and UniLectin are suitable connectors between virology, protein science, and chemical biology. This of course reflects the biological role of lectins as key players in cell–cell communication. Since SugarBindDB is dedicated to the characterization of pathogen lectins, it is naturally connected to ViralZone. Unilectin is by definition universal and collects data that cover all kingdoms. Since the intricate network of interactions at the host cell surface also involves host lectins recognizing pathogen glycans, a full description of the situation may require further data from UniLectin.

UniLectin includes UniLectin3D, which unsurprisingly compiles three-dimensional information regarding lectins and glycan ligands. In particular, some structures available in this database provide details of viruses binding to glycans of animal cell surfaces (rotavirus, influenza, etc.) and to bacterial cell walls (bacteriophage). In Norovirus, the capsid protein sequences differ among the several genotypes, resulting in a fine specificity for different human histo-blood groups. Figure 5 illustrates the tree-search for noroviruses together with selected structures of GI-7 genotype in complex with Lewis Y and blood group A tetrasaccharides as described in [52]. These screenshots also highlight the links with external databases, as well as the details of the cited publication that are available through the green box expansion.

**Figure 5 viruses-11-00374-f005:**
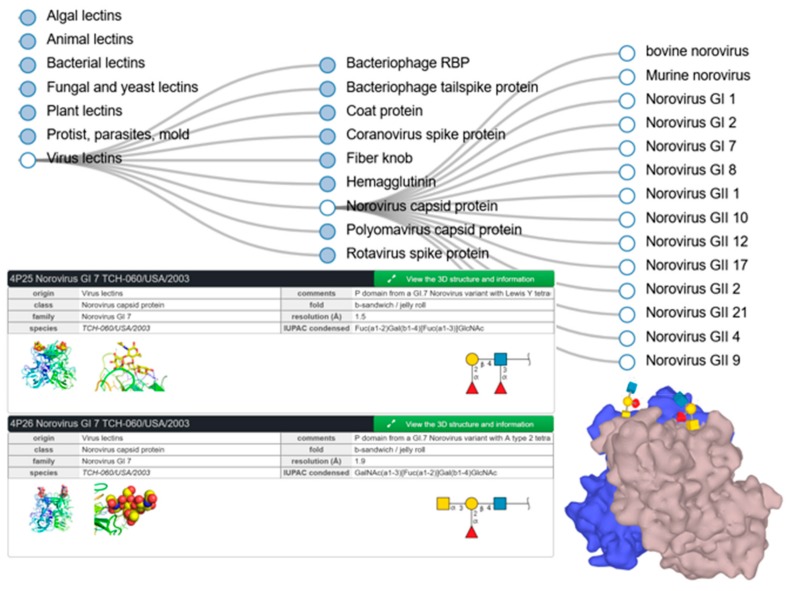
Partial result of UniLectin3D search for Norovirus, with selected examples of entries of GI-7 genotype.

As a final comment in this section, note that new virus-related entries in SugarBindDB are very slowly finding their way into the database due to the poor funding of biocuration. This raises the question of associating experts with the database development and is hardly a hidden invitation.

## 4. Discussion

Bioinformatics tools have become indispensable for processing and analyzing the now popular -omics data. Raw data associated with published articles are routinely submitted in repositories, while dedicated databases are developed to shape the implicit knowledge revealed in analyzed -omics data. Such a setup is now operational to support the enlightened interpretation of new results in genomics, transcriptomics, and proteomics, as well as to maximize data sharing, reuse, and integration. Based on this interoperability, a broad range of automated procedures are implemented to collect and piece together information from databases and support the formulation of new and testable hypotheses. At this stage, glycomics has not quite yet reached that level despite the essential roles of glycans in the mediation of molecular interactions. In the particular case of virology, this lack is visible.

The panoramic view of the different information resources covering interactions between virus and glycans as shown in Figure 2, should help to improve both the breadth and depth of knowledge. Obviously, the current connectivity between resources is far from being complete and new cross-links are needed for users to make the most of all data disseminated in several resources. We strive to fill that gap. As partially mentioned earlier, near future plans encompass new content in MatrixDB that will enable the creation of cross-links between SugarBindDB and MatrixDB, with a focus on GAG-binding pathogens. Furthermore, in order to make the substrates and products of enzymatic reactions collected in CAZy more explicit, substantial work on harmonizing knowledge representation is underway. This requires the inclusion of information from the Rhea database to generate the association of each known glycan structure of GlyConnect with its required enzyme toolbox. The main obstacle to this apparently straightforward application lies in the limitations in format sharing between chemical biology, carbohydrate chemistry, and glycobiology. Figure 6 illustrates the situation by highlighting format (in)compatibility between databases. The diversity of information sources is yet again creating barriers of communication. Standards such as SMILES (Simplified Molecular Input Line Entry Specification) and InChi (International Chemical Identifier) have been adopted in chemical biology and cheminformatics, while IUPAC (International Union of Pure and Applied Chemistry) remains the gold reference in biochemistry. Glycobiology has developed a series of encoding formats that were attempts in a variety of ways to account for the branched structures. In the end, the multiplicity is such that toolboxes of converters have been developed over the years [53,54]. Nonetheless, the (glyco-)bioinformatics community has opted for GlycoCT as a structure encoding format [55] shared by the majority of databases. Furthermore, it seems that SNFG [41] is spreading as a unique code for representing glycans. Despite growing consensus in each community for relying on shared standards, cross-talk is not established between disciplines, and requires the bridging tasks we have undertaken.

**Figure 6 viruses-11-00374-f006:**
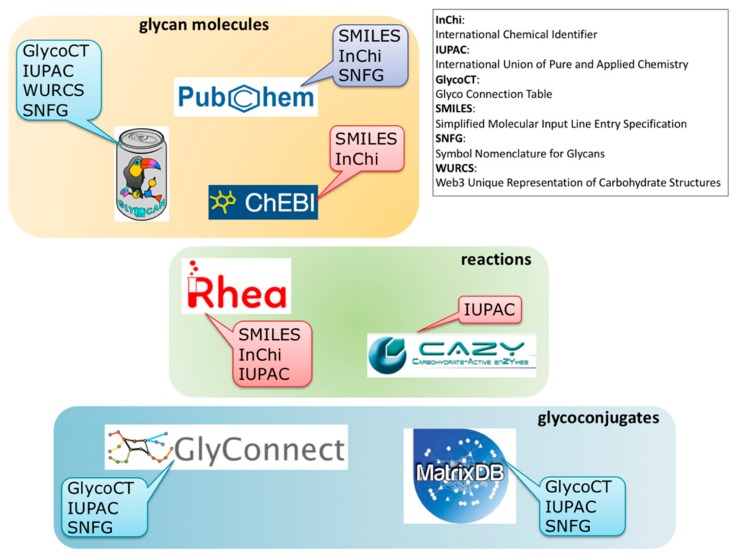
Variety of (in)compatible molecule and reaction encoding formats associated with each database in each category (molecules/reactions/glycoconjugates) hindering smooth cross-referencing.

Moreover, there are missing data that would be very useful for virology. For example, tissue localization of various glycans in vertebrates would be invaluable to correlate with virus tissue infections. This kind of resource exists for proteins [56], but not yet for glycans.

Finally, a paradigm shift is also needed to advance the field of glycovirology. For example, a comprehensive survey of known glycosylation in *Mononegavirales* was recently published [57]. Had the authors known about ViralZone, the valuable though PDF-static information printed in this article could have been structured in a way to be stored straightaway in the database and enhance virus annotation. It takes bioinformaticians days of work to search the relevant information often hidden in supplementary material of published articles when simple awareness of reference online resources could suffice to motivate authors to disseminate their work through interacting upstream with database developers. For as long as automated data submission procedures are not imposed as they are now for DNA sequence or protein mass spectrometry data, glyco-viro-database improvement relies on researchers’ good will to actively share their data. With the present article, we wish to convey the advantages of increasing bioinformatics usage in glyco-virology and show how it reveals implicit correlations through searching cross-referenced resources. The more complete the stored data, the better the chances of unearthing gems.

## 5. Conclusions

Glycans are prevalent molecules that offer a broad range of interplay with viruses crossing the extracellular space to infect new cells. Bioinformatics resources are critical for the analysis of the diversity of available data. We have provided a broad overview of the bioinformatics landscape of glycan–virus interactions. This has fed a reflection and discussion on the need and the means of increasing interoperability between existing databases as a key to support interaction data interpretation. The next step is surely to focus on bridging more tightly immunology and glyco-virology resources.

## Figures and Tables

**Figure 1 viruses-11-00374-f001:**
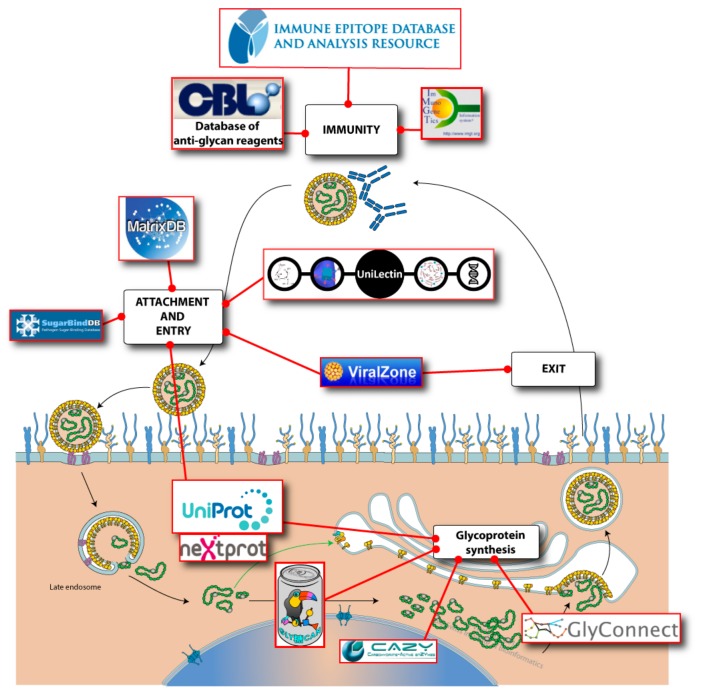
Virus interaction with glycans mapped with corresponding bioinformatics resources.

**Figure 2 viruses-11-00374-f002:**
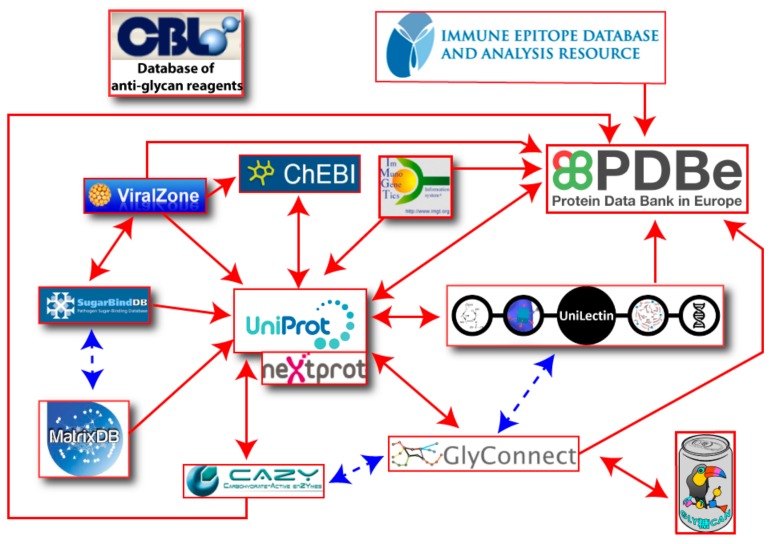
One- or two-way referencing between online databases. Arrows represent links between different resources and are double edged if links are reciprocal. Blue dotted lines represent future links that will strengthen the framework of databases.

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
