# Peer review of "A Bioinformatics View of Glycan–Virus Interactions"

_viruses, 2019, doi:10.3390/v11040374_

Reviewer 1 Report

1. Lack a critical review of each database about the quality, coverage, usefulness, etc.

2. Need further proofreading

Author Response

Given the consistent comments of both reviewers regarding the description of databases we have added a large table itemising each resource and providing descriptive details as well as examples of use. We also inserted a smaller table for software.

We have also proof read the manuscript as requested.

Reviewer 2 Report

The article presented is a review article titled "A bioinformatics view of glycan-virus interactions", which compiles a catalog of bioinformatics tools that can correlate glycomics with viromics. The review is well researched, comprehensive and would greatly benefit the field of virology, but at the same time provide essential feedback to informaticians to develop new technologies which could fill in gaps in the current state of the field. 

The following minor points can be addressed to improve the quality of the manuscript prior to publication:

1. The first paragraph of the introduction needs some citations for the average reader. Not all scientists would appreciate the surface abundance of glycans and their effects to viral entry. References here would add weight to the claims being made and provide direction to the readers if they need to check other information.

2. The sentence "This effect is referred as to the glycan shield and the best example of this feature is the HIV virus, known to have the highest density of glycans attached to its surface proteins [7][8]." makes it feel that reference 8 pertains to the high density of glycans on HIV virus. However reference 8 is about Lassa virus. The authors should consider rephrasing the sentence to expand the scope for e.g. "This effect is referred to as the glycan shield, and the best example of this feature are the HIV virus (known to have the highest density of glycans attached to its surface proteins) and the Lassa virus [7][8]."

3. The authors make a statement "We also show that integrating complementary information from multiple sources is not straightforward." While this is true, several efforts are being undertaken to integrate such databases together. For example, the authors seem to be unaware of GlyGen initiative (glygen.org) which aims to unify such data with respect to "glycomics, genomics, proteomics (and glycoproteomics), cell biology, developmental biology and biochemistry". Appreciating such efforts and including them in the manuscript would be useful. Also rephrasing the sentence to include a sentence like "Efforts to integrate databases are being made, but there are several opportunities to enhance such cohesiveness" should be considered.

4.  Figure 1 legend should list the different databases so that it is better indexable by search engines.

5. A major improvement would be to have a summary table of tools with the Tool Name, Web Address, Use, Example Use Case for Virology, Reference. Such a table would guide the readers of the journal to how they could implement these tools into their workflow to assist them to making better decisions.

6. Apart from the problems associated with the tools higlighted by the authors in the field, there is an additional factor of reliability and quality of data. For example GlyConnect at the time of this review seemed to give a "Service Temporarily Unvailable" error when trying to use their "Browse" feature. Another example is GlyToucan's IUPAC names seems to be broken (the names for complex N-glycans for example are truncated). It would be impossible for scientists who do chemistry or biology on the bench to come up with WURCs or GlycoCT nomenclature for a glycan of interest which they either synthesized, identified or found in a publication. On the other hand, chemists and biologists find it easier to write names as linear IUPAC or similar names. Another example, with SugarBind if you try to use the Glycan Builder tool, it gives an alert stating "Failed to load the bootstrap JavaScript: VAADIN/vaadinBootstrap.js", and does not load any interface to build the glycan.Such issues with the current tools makes it impossible for scientists to use these tools reliably. UniProtKB is well established and used regularly due to the reliability of these services. 

While I understand it is beyond the control of the authors to control all these tools, it should be highlighted that work needs to be done in order to improve (a) the quality of information being provided (b) the reliability of service being provided.

7. The authors state "characterised in PubMed ID 12825167." should provide the reference to the paper.

8. The Norovirus example should be placed in a sub heading of its own to let readers know that this is an example walkthrough of an analysis.

9. The authors state "a new platform is briefly introduced in a just released Application Note [35]." without giving any clue to the readers what it is. It would be useful to describe what is the new platform in a sentence.

10. Sentence "UniProtKB is also a most reliable source for describing host glycoproteins..." grammar is wrong, please correct this.

11. The authors give an example of SugarBindDB entry for "Norovirus Norwalk virus GV", however, I could not find such a virus. There was however a Norovirus norwalk virus GrV which has the information. I suppose this was a typo which can be easily corrected, but recommend the authors to just check if this was the case.

Overall, the paper is of good quality and great importance to the field of virology and provides a guidance to researchers on tools available to study viruses. I recommend this manuscript for publication once the above minor points are addressed.

Author Response

 1. The first paragraph of the introduction needs some citations for the average reader. Not all scientists would appreciate the surface abundance of glycans and their effects to viral entry. References here would add weight to the claims being made and provide direction to the readers if they need to check other information.

 We have added 5 references in the first and 1 in the last paragraphs of the introduction.

 2. The sentence "This effect is referred as to the glycan shield and the best example of this feature is the HIV virus, known to have the highest density of glycans attached to its surface proteins [7][8]." makes it feel that reference 8 pertains to the high density of glycans on HIV virus. However reference 8 is about Lassa virus. The authors should consider rephrasing the sentence to expand the scope for e.g. "This effect is referred to as the glycan shield, and the best example of this feature are the HIV virus (known to have the highest density of glycans attached to its surface proteins) and the Lassa virus [7][8]."

We thank the reviewer for this relevant comment and have changed the text accordingly

3. The authors make a statement "We also show that integrating complementary information from multiple sources is not straightforward." While this is true, several efforts are being undertaken to integrate such databases together. For example, the authors seem to be unaware of GlyGen initiative (glygen.org) which aims to unify such data with respect to "glycomics, genomics, proteomics (and glycoproteomics), cell biology, developmental biology and biochemistry". Appreciating such efforts and including them in the manuscript would be useful. Also rephrasing the sentence to include a sentence like "Efforts to integrate databases are being made, but there are several opportunities to enhance such cohesiveness" should be considered.

We are certainly aware of the GlyGen initiative however, as far as we know (as described in the website) it is focused on integrating human and mouse data. This is the reason why it was not cited though we would have gladly referred to it, had we been reviewing mammalian-related data integration. We describe here mainly virology-based data integration and this is highlighted in the first two figures. Furthermore, we are showing existing implementation in the virus-focused ecosystem of ViralZone, which is another reason why GlyGen was not included. It is not present in this context.

4.  Figure 1 legend should list the different databases so that it is better indexable by search engines.

5. A major improvement would be to have a summary table of tools with the Tool Name, Web Address, Use, Example Use Case for Virology, Reference. Such a table would guide the readers of the journal to how they could implement these tools into their workflow to assist them to making better decisions.

We addressed a related comment of Reviewer1 and these two comments by including summary tables that provide the requested details of all cited databases (Table1) and software (Table2).

6. Apart from the problems associated with the tools higlighted by the authors in the field, there is an additional factor of reliability and quality of data. For example GlyConnect at the time of this review seemed to give a "Service Temporarily Unvailable" error when trying to use their "Browse" feature. Another example is GlyToucan's IUPAC names seems to be broken (the names for complex N-glycans for example are truncated). It would be impossible for scientists who do chemistry or biology on the bench to come up with WURCs or GlycoCT nomenclature for a glycan of interest which they either synthesized, identified or found in a publication. On the other hand, chemists and biologists find it easier to write names as linear IUPAC or similar names. Another example, with SugarBind if you try to use the Glycan Builder tool, it gives an alert stating "Failed to load the bootstrap JavaScript: VAADIN/vaadinBootstrap.js", and does not load any interface to build the glycan.Such issues with the current tools makes it impossible for scientists to use these tools reliably. UniProtKB is well established and used regularly due to the reliability of these services.

We are sorry the reviewer was so unlucky at the time of testing resources. Thankfully “temporarily” means a short period of time and this hopefully invites users to another trial that is usually more successful. UniProt is indeed well established and benefits from over 30 years of experience. GlyTouCan and GlyConnect were released in the past two years and do suffer from teething problems. However, their respective developers are aware that the service can be improved and are actively working at it. We also make sure that ViralZone is another reliable service given its central role in this manuscript. By including creation dates in Table 1 we highlight the potential differences in stability between databases. This comment is included in the text.

While I understand it is beyond the control of the authors to control all these tools, it should be highlighted that work needs to be done in order to improve (a) the quality of information being provided (b) the reliability of service being provided.

Indeed, due to the extended workload in developing glycoinformatics tools, the community relies on existing resources in an attempt not to reinvent the wheel and this is the case for GlycanBuilder. It was developed by another group and was the best option for drawing glycans a few years ago. Note that there is a cautionary message on our websites using it, explaining that this tool was imported (“The graphical interface is not part of the software maintained by SIB. We apologize if something goes wrong with it.”). This shows that we are acutely aware of the problem and would have liked to solve all issues simultaneously but it is not realistic as acknowledged by the reviewer. We are in a transition phase while accounting for this shortcoming and have recently published a prototype named SugarSketcher designed as an open-source tool.

The information requested is in the table mentioned in the previous answer.

7. The authors state "characterised in PubMed ID 12825167." should provide the reference to the paper.

 We apologise for this oversight and have now included the reference as it was in line 110.

8. The Norovirus example should be placed in a sub heading of its own to let readers know that this is an example walkthrough of an analysis.

 We thank the reviewer for this relevant comment and have changed the headers accordingly:

2.1 Overview, 2.2 Illustration of usage

9. The authors state "a new platform is briefly introduced in a just released Application Note [35]." without giving any clue to the readers what it is. It would be useful to describe what is the new platform in a sentence.

It was very contextual but to be as clear as possible, we added the part in bold:

“At this point in time, US-based glycan array data collecting and mining is in transition between old and new initiatives. As we write, a new platform designed to visualize, analyze and compare glycan microarray data is briefly introduced in a just released Application Note [35]. This shows that bioinformatics applied to processing and analysing glycan array data is still very much in the making.”

10. Sentence "UniProtKB is also a most reliable source for describing host glycoproteins..." grammar is wrong, please correct this.

“Most” was removed in this sentence.

11. The authors give an example of SugarBindDB entry for "Norovirus Norwalk virus GV", however, I could not find such a virus. There was however a Norovirus norwalk virus GrV which has the information. I suppose this was a typo which can be easily corrected, but recommend the authors to just check if this was the case.

This is the result of non-conclusive discussion we had between co-authors regarding “official” naming (GV) vs “name provided in associated published article” (GrV). We apologise for not sorting it out in the final version. We made this now clear in the text by adding:

“the official designation should be GV according to ViralZone but SugarBindDB reports the name found in the cited source”

Round  2

Reviewer 1 Report

Significant editing efforts have made the manuscript more readable.